# Sim-to-Lab-to-Real: Safe Reinforcement Learning with Shielding and Generalization Guarantees

**Kai-Chieh Hsu**[*]  **Allen Z. Ren**[*]  **Duy P. Nguyen**  **Anirudha Majumdar**[†]  **Jaime F. Fisac**[†]

Princeton University
{kaichieh, allen.ren, duyn, ani.majumdar, jfisac}@princeton.edu

## Abstract

Safety is a critical component of autonomous systems and remains a challenge for learning-based policies to be utilized in the real world. In this paper, we propose Sim-to-Lab-to-Real to bridge the reality gap with a probabilistically guaranteed safety-aware policy distribution. To improve safety, we apply a dual policy setup where a performance policy is trained using the cumulative task reward and a backup (safety) policy is trained by solving the safety Bellman Equation based on Hamilton-Jacobi reachability analysis. In *Sim-to-Lab* transfer, we apply a supervisory control scheme to shield unsafe actions during exploration; in *Lab-to-Real* transfer, we leverage the Probably Approximately Correct (PAC)-Bayes framework to provide lower bounds on the expected performance and safety of policies in unseen environments. We empirically study the proposed framework for ego-vision navigation in two types of indoor environments including a photo-realistic one. We also demonstrate strong generalization performance through hardware experiments in real indoor spaces with a quadrupedal robot.[3]

## 1  Introduction

Due to tight hardware constraints and high sample complexities, reinforcement learning with robots is usually performed solely in simulated environments. However, robots' performance often degrades sharply in the real world. *Domain randomization* has helped bridge this *Sim-to-Real* gap by simulating a wide range of scenarios [1, 2], but does not explicitly address *safety* of the robots. Although training in simulation allows safety violations, without training to avoid unsafe behavior, robots tend to exhibit similarly unsafe behavior in real environments. Another drawback of these techniques is that they do not provide any *generalization guarantee* on robots' performance or safety to different real environments, which is necessary for deploying autonomous systems in safety-critical scenarios (e.g., households with children).

In this work, we explore a middle-level training stage between *Sim* and *Real*, which we call *Lab*, that aims to further bridge the *Sim-to-Real* gap. The proposed *Sim-to-Lab-to-Real* framework is motivated by the conventional engineering practice that before deploying autonomous systems in the real world after training, human designers usually test systems in a more realistic but controlled environment, such as a test track for autonomous cars. Our intuition is that (1) after training in diverse conditions in simulation, the robot *fine-tunes* in more specific environments before deployment in similar environments in the real world; (2) this second stage also provides *guarantees* on the performance and safety of the system in *Real* deployment. Fig. 1 shows the pipeline.

---

[*]Equal contributions in alphabetical order
[†]Equal contributions in advising
[3]See https://sites.google.com/princeton.edu/sim-to-lab-to-real for representative trials.

36th Conference on Neural Information Processing Systems (NeurIPS 2022).

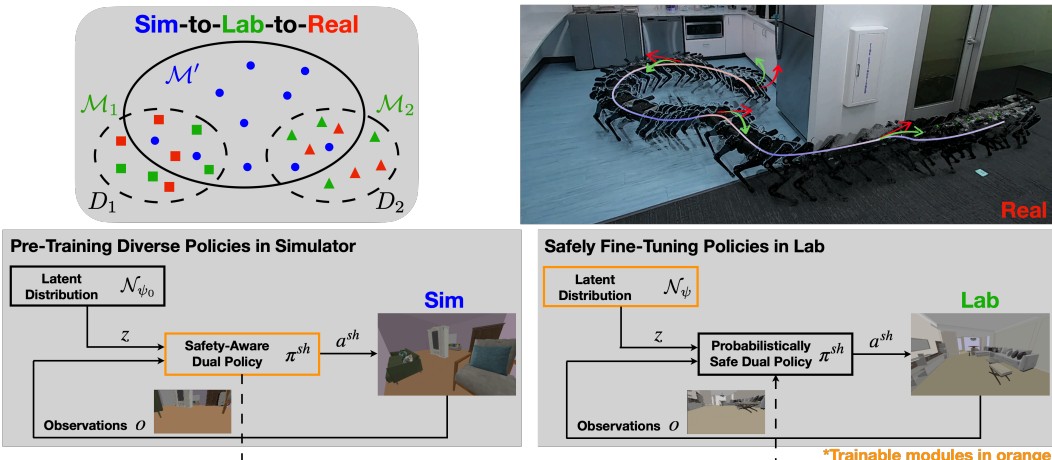

Figure 1: **Overview of the Sim-to-Lab-to-Real framework.** *Sim* stage trains a latent-conditioned, safety-aware dual policy in a wide variety of conditions. Then the *Lab* stage safely fine-tunes the latent distribution in different and more specific settings, which are also closer to Real environments.

In the *Lab* training, the autonomous system needs to explore safely to further improve the performance. Our approach builds upon a dual policy setup where a performance policy optimizes task reward and a backup (safety) policy keeps robots away from unsafe regions. We then apply a *least-restrictive control law* (or *shielding*) [3]: the backup policy only intervenes when the safety state-action value function deems the proposed action from performance policy violates safety constraints in the future. The backup policy is pre-trained in the *Sim* stage and ready to ensure safe exploration once *Lab* training starts. Based on safe RL training using *Hamilton-Jacobi (HJ) reachability-analysis* developed in [4, 5], our backup agent can learn from *near failure* with *dense* signals; even when the backup policy updates itself in safety-critical conditions, the training does not rely on safety violations unlike previous work that uses binary safety indicators [6, 7].

We also apply the *Probably Approximately Correct (PAC)-Bayes Control* framework [8] to provide bounds on the expected performance and safety in unseen environments. The framework fits our setup as its two training stages, *prior* and *posterior*, correspond to *Sim* and *Lab*. We train a *distribution of policies* by conditioning the performance policy on latent variables sampled from a distribution. After training a *prior* distribution in *Sim* stage, we fine-tune it in *Lab* and obtain a *posterior* policy distribution and its associated generalization guarantee. Unlike other techniques from robust control [9] and reachability analysis [10], PAC-Bayes Control does not assume knowledge of the uncertainty affecting the system (e.g., bound on actuation noise) or the environment (e.g., minimum distance between obstacles), and allow training policies with rich sensing like vision.

## 2    Related Work

**Safe Exploration**    Recent methods [6, 7, 11, 12] address safe exploration in training with similar shielding schemes as in this work. However, the major differences lie in how the safety state-action value function, *safety critic*, is trained and how the backup action is generated. Previous work learn the safety critic from only sparse (binary) safety labels. Srinivasan et al. [6] use this critic to filter out *unsafe* actions until the performance policy resamples a safe one, while Thananjeyan et al. [7] train the same critic but use action from the backup policy instead of resampling the performance policy. One concurrent work [12] uses the same reachability-based RL to learn the backup agent. Our method is distinct in that we propose the two-stage training to allow safer exploration and train the reachability-based RL end-to-end from images without pre-training the visual encoder.

**Generalization Theory and Guarantees**    In supervised learning, generalization theory provides guarantees on the expected loss on new samples drawn from the unknown data distribution, after training a model using a finite number of samples. Recent work based on PAC-Bayes generalization theory [13] have provided non-vacuous bounds for neural networks in supervised learning [14]. [8] apply the PAC-Bayes framework in policy learning settings and provide generalization guarantees for control policies in unseen environments. Follow-up work has provided strong guarantees in different robotics settings including for learning neural network policies for vision-based control [15, 16, 17].

However, previous work has not adopted safety-related policy architectures nor considered safety *during training*.

**Safe Visual Navigation in Unseen Environments**    Typical approaches in robot navigation focus on explicit mapping of the environment combined with long-horizon planning [18, 19]. Recently there has been a line of work in applying Hamilton-Jacobi reachability analysis in visual navigation to improve the safety of the agent. [20] solve for the reachability set at each step but relies on a map generated using onboard camera. [21] propose supervising the visual policy using expert data generated by solving the reachability set. Our work also leverages reachability analysis but does not build a map of the environment nor relies on offline data generated by a different (expert) agent.

## 3    Problem Formulation and Preliminaries

We consider a robot with discrete-time dynamics given by $s_{t+1} = f_E(s_t, a_t)$ with state $s \in \mathcal{S} \subseteq \mathbb{R}^{n_s}$, control input $a \in \mathcal{A} \subseteq \mathbb{R}^{n_a}$, and environment $E \in \mathcal{E}$ that the robot interacts with (e.g., an indoor space with furniture including initial and goal locations of the robot). Below we introduce the different conditions of the environments considered in the three stages.

**Environment - Sim.** In the Sim stage, we assume there is a set of training environments $\mathcal{M}' \subset \mathcal{E}$ (e.g., synthetic indoor spaces with randomized arrangement of furniture), $\mathcal{M}' := \{E_1, E_2, \cdots, E_{N'}\}$. There is no assumption on how $\mathcal{M}'$ is distributed in $\mathcal{E}$.

**Environment - Lab.** In the Lab stage, we are concerned with more specific conditions, and there can be different distributions of environments $D_1, D_2, ...$ (e.g., office or home spaces, dimensions of the obstacles), with which the policies trained in Sim can be fine-tuned. We assume *no* explicit knowledge of each distribution $D_i$; instead, we assume there is a set of $N_i$ training environments drawn i.i.d. from $D_i$ available for the robot to train in; we denote these training datasets by $\mathcal{M}_i := \{E_1, E_2, \cdots, E_{N_i}\} \sim D_i^{N_i}$. With a slight abuse of notations and for convenience, we consider a single distribution when introducing the rest of formulation and the approach, and we denote the concerned distribution $D$, the training set $\mathcal{M}$, and the number of training environments $N$.

**Environment - Real.** In the Real stage, we assume the robot is deployed in environments from the same distribution $D$ but *unseen* during the Lab stage.

Next we introduce the rest of problem settings including the robot sensor, the policy, and robot's task involving the reward function and the failure set. These settings hold the same for all three stages, except for the failure set which we do not require knowledge of at Real deployment.

**Sensor.** In all environments, we assume the robot has a sensor (e.g., RGB camera) that provides an observation $o = h_E(s)$ using a sensor mapping $h : \mathcal{S} \times \mathcal{E} \rightarrow \mathcal{O}$.

**Task and Policy.** Suppose the robot's task can be defined by a reward function, and let $R_E(\pi)$ denote the cumulative reward gained over a (finite) time horizon by a deterministic policy $\pi : \mathcal{O} \rightarrow \mathcal{A}$ when deployed in an environment $E$. We assume the policy $\pi$ belongs to a space $\Pi$ of policies. We also allow policies that map *histories* of observations to actions by augmenting the observation space to keep track of observation sequences. We assume $R_E(\pi) \in [0, 1]$, but make no further assumptions such as continuity or smoothness. We use $\xi_E^{s,\pi} : [0, T] \times \mathcal{E} \rightarrow \mathcal{S}$ to denote the trajectory rollout from state $s$ using policy $\pi$ in the environment $E$ up to a time horizon $T$.

**Failure set.** We further assume there are environment-dependent failure sets $\mathcal{F}_E \subseteq \mathcal{S}$, that the robot is not allowed to enter. In training stages, we assume the robot has access to Lipschitz functions $g : \mathcal{S} \times \mathcal{E} \rightarrow \mathbb{R}$ such that $\mathcal{F}_E$ is equal to the zero superlevel set of $g_E$, namely, $s \in \mathcal{F}_E \Leftrightarrow g_E(s) \geq 0$. For example, $g_E(s)$ can be the signed distance function to the nearest obstacle around state $s$. Thus, $g_E(s)$ is called the safety margin function throughout the paper.

### 3.1   Goal

Our goal is to learn policies that *provably generalize* to unseen environments at the Real stage. This is very challenging since we do not have explicit knowledge of the underlying distribution $D$. We employ a slightly more general formulation where a *distribution $P$ over policies* $\pi \in \Pi$ instead of a single policy is used. In addition to maximizing the policy reward, we want to minimize the number of safety violations, i.e., the number of times that the robot enters failure sets. Our goal can then

be formalized by the following optimization problem, which we would like to lower bound as the guarantee:

$$R^\star := \sup_{P \in \mathcal{P}} R_D(P), \text{ where } R_D(P) := \mathop{\mathbb{E}}_{E \sim D} \left[ \mathop{\mathbb{E}}_{\pi \sim P} \left[ R_E(\pi) \right] \right], \tag{1}$$

$$R_E(\pi) := \overline{R}_E(\pi) \mathbb{1} \left\{ \forall t \in [0, T], \xi_E^{s,\pi}(t) \notin \mathcal{F}_E \right\}, \tag{2}$$

where $\overline{R}_E(\pi) \in [0, 1]$ denotes the task reward that does not penalize safety violation, $\mathcal{P}$ denotes the space of probability distributions on the policy space $\Pi$, and $\mathbb{1}\{\cdot\}$ is the indicator function. Here the task reward can be either dense (e.g., normalized cumulative reward) or sparse (e.g., reaching the target or not).

## 3.2 Generalization Bounds

Recently, PAC-Bayes generalization bounds have been applied to policy learning settings in order to provide formal generalization guarantees in unseen environments. We briefly introduce this framework here, as it will be used in our overall approach presented in Section 4. First it requires training a prior policy distribution $P_0$, which we do in the Sim stage with the set of environments $M'$. Then in the Lab stage, we fine-tune $P_0$ with environments $M$ to obtain the posterior distribution $P$. Now, define the *empirical reward* of $P$ as the average expected reward across training environments in $\mathcal{M}$:

$$R_{\mathcal{M}}(P) := \frac{1}{N} \sum_{E \in \mathcal{M}} \mathop{\mathbb{E}}_{\pi \sim P} \left[ R_E(\pi) \right]. \tag{3}$$

The following theorem can then be used to lower bound the true expected reward $R_D(P)$.

**Theorem 1** (PAC-Bayes Bound for Control Policies; adapted from [8]). *Let $P_0 \in \mathcal{P}$ be a prior distribution. Then, for any $P \in \mathcal{P}$, and any $\delta \in (0, 1)$, with probability at least $1 - \delta$ over sampled environments $\mathcal{M} \sim D^N$, the following inequality holds:*

$$R_D(P) \geq R_{\text{PAC}}(P, P_0) := R_{\mathcal{M}}(P) - \sqrt{C(P, P_0)}, \text{ where } C(P, P_0) := \frac{\text{KL}(P\|P_0) + \log(\frac{2\sqrt{N}}{\delta})}{2N},$$

*and $\text{KL}(P\|Q)$ stands for Kullback-Leibler divergence between probability distribution $P$ and $Q$.*

Maximizing the lower bound $R_{\text{PAC}}$ can be viewed as maximizing the empirical reward $R_{\mathcal{M}}(P)$ along with minimizing a regularizer $C$ that prevents overfitting by penalizing the deviation of the posterior $P$ from the prior $P_0$. By fine-tuning $P_0$ to $P$ and maximizing the bound in the Lab stage, we can provide a generalization guarantee for trained policies in unseen environments in the Real stage.

## 4 Method

Our proposed *Sim-to-Lab-to-Real* framework bridges the reality gap with probabilistic guarantees by learning a safety-aware policy distribution. Fig. 2 shows the architecture of the safety-aware policy distribution. It explicitly handles safety by leveraging a shielding classifier, which monitors the candidate actions from the performance policy and replaces them with backup actions when necessary. We also condition the performance policy on a latent variable to encode diverse strategies. We show how to jointly train a dual policy conditioning on a latent distribution in Sec. 4.1 (*Sim-to-Lab*). The details of *Lab* training and derivations of generalization guarantees are provided in Sec. 4.2 (*Lab-to-Real*).

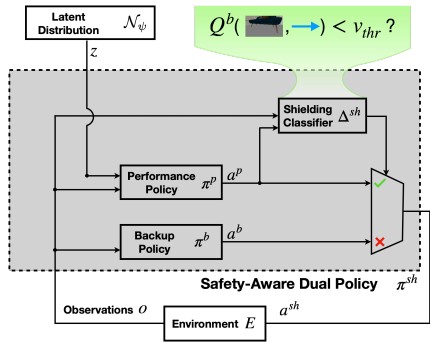

Figure 2: Architecture of the safety-aware policy distribution.

For training, we use a proxy reward function $r_E : \mathcal{S} \times \mathcal{A} \times \mathcal{E} \to \mathbb{R}$ (e.g., dense reward in distance to target) as a single-step surrogate for the task reward $\overline{R}_E(\pi)$. At each step the robot also receives a safety cost $g_E(s)$ (e.g., distance to nearest obstacle). We train the dual policy with modifications of the Soft Actor-Critic (SAC) algorithm [22]. We denote the neural network (NN) weights of the actor and critic $\theta$ and $w$. We use superscripts $(\cdot)^{\text{p}}$ and $(\cdot)^{\text{b}}$ to denote critics, actors, and actions from

the performance or backup agent. The performance policy is conditioned on latent variable $z \in \mathbb{R}^{n_z}$ sampled from a multivariate Gaussian distribution with diagonal covariance as $z \sim \mathcal{N}(\mu, \Sigma)$, where $\mu \in \mathbb{R}^{n_z}$ is the mean and $\Sigma \in \mathbb{R}^{n_z \times n_z}$ is the diagonal covariance matrix. We further denote $\sigma \in \mathbb{R}^{n_z}$ the element-wise square-root of the diagonal of $\Sigma$, and define $\psi = (\mu, \sigma)$, $\mathcal{N}_\psi := \mathcal{N}(\mu, \text{diag}(\sigma^2))$.

## 4.1 Pre-Training Diverse Dual Policy in Simulator

In this Sim stage, we use the dataset $\mathcal{M}'$ that contains environments that are not necessarily similar to those from the target environment distribution $D$. They contain randomized properties such as random arrangement of furniture in indoor space and random camera tilting angle on the robot.

**Safety through Reachability-Based Reinforcement Learning** Failures are usually catastrophic in safety-critical settings; thus worst-case safety, instead of an average safety over the trajectory, should be considered. For training the backup policy, we incorporate reachability-based reinforcement learning [4] and optimize the discounted safety Bellman equation (DSBE):

$$Q^{\text{b}}(o_t, a_t) := (1 - \gamma)g_E(s_t) + \gamma \max \left\{ g_E(s_t), \min_{a_{t+1} \in \mathcal{A}} Q^{\text{b}}(o_{t+1}, a_{t+1}) \right\}, \quad (4)$$

where $o_t = h_E(s_t)$ and $\gamma$ is the discount factor. This discount factor represents how much the RL agent cares about future outcomes: if $\gamma$ is small, the RL agent is myopic and only cares about the current "danger", and as $\gamma \to 1$, it recovers the infinite-horizon safety state-action value function. In training, we initialize $\gamma = 0.8$ and gradually anneal $\gamma \to 1$ as the backup policy improves.

The safety critic in (4) captures the maximum cost $g_E$ along the trajectory starting from $s_t$ with action $a_t$ even if the best control input is applied at every instant afterward. Thus, $\min_{a_t \in \mathcal{A}} Q(o_t, a_t) > 0$ indicates the robot is predicted to hit an obstacle in the future. DSBE allows the backup agent to learn the safety critic from near failure, which significantly reduces failure events during training. DSBE also updates the backup agent with dense signals, which is more suitable for the joint training of performance and backup agents.

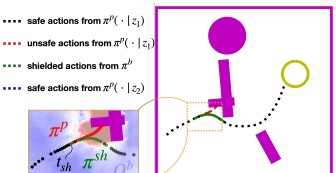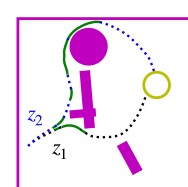

Figure 3: Safe and diverse trajectories generated by the safety-aware policy distribution. The inset shows safety values $Q(o, \pi^{\text{b}}(o))$ with the observation $o$ taken when the heading angle fixed to the one at time instant $t_{sh}$.

**Shielding** We leverage shielding to reduce the number of safety violations in both training and deployment. Besides the backup policy $\pi^{\text{b}}$, we also train a performance policy $\pi^{\text{p}}$ to maximize task reward. Before a candidate action from the performance policy is applied, a shielding classifier $\Delta^{\text{sh}}$ checks if it is safe. We replace it with the action from the backup policy if and only if that candidate action is predicted to cause safety violation in the future. The shielding criterion is summarized in (5). This ensures minimum intervention by the backup policy while the performance policy guides the robot towards the target [3, 23].

$$\pi^{\text{sh}}(o) = \begin{cases} \pi^{\text{p}}(o), & \Delta^{\text{sh}}(o, \pi^{\text{p}}, \pi^{\text{b}}) = 1 \\ \pi^{\text{b}}(o), & \text{otherwise} \end{cases}. \quad (5)$$

The safety value function learned by DSBE represents the maximum cost along the trajectory in the future if the learned policy is followed. Based on this, we propose *Value-based Shielding* with a physically meaningful shielding threshold, i.e., it represents the margin to failure. Once the robot receives the current observation $o$ and uses performance policy to generate action $a^{\text{p}}$, the backup policy overrides the action if and only if $Q^{\text{b}}(o, a^{\text{p}}) > v_{thr}$. In other words, the shielding discriminator is defined as below

$$\Delta^{\text{sh}}(o, \pi^{\text{p}}, Q^{\text{b}}) := \mathbb{1}\left\{ Q^{\text{b}}(o, \pi^{\text{p}}(o)) \le v_{thr} \right\}. \quad (6)$$

Fig. 3 shows an example of shielding that prevents applying unsafe actions from the performance policy (with shielding, the red dotted lines are replaced with green dotted lines in the inset). We compare the safety critic based on DSBE with ones learned with sparse safety indicators [6, 7] in Sec. 5 and Fig. 6; our approach affords much better safety during training and deployment.

**Joint Training of Dual Policy.** In Sim stage, we fix the latent distribution to be a zero-mean Gaussian distribution with diagonal covariance $\mathcal{N}_{\psi_0}$, where $\psi_0 = (0, \sigma_0)$. For each episode during training, we sample a latent variable $z \sim \mathcal{N}_{\psi_0}$ and condition the performance policy on it for the whole episode. The training procedure is illustrated in Algorithm 1.

Since we train both policies with modifications of the off-policy SAC algorithm, we can use transitions with actions proposed by either policy. The transitions are stored in a shared replay buffer. At every step during training, the robot needs to select a policy to follow. We introduce $\rho$, the probability that the robot chooses an action from the backup policy. We initialize $\rho$ to 1, which means initially all actions are sampled from the backup policy. Intuitively, the backup policy needs to be trained well before shielding is used in training. We gradually anneal $\rho \to 0$. We then introduce another parameter $\epsilon$, the probability that the shielding is activated at the step. This parameter represents how much the backup policy is trusted to shield the performance policy. We typically anneal $\epsilon$ from 0 to 1. The influence of $\rho$ and $\epsilon$ are further analyzed in Appendix A.5.

---

**Algorithm 1** Joint training in simulator

**Require:** $\mathcal{M}', \pi^p, \pi^b, \mathcal{N}_{\psi_0} := \mathcal{N}(0, \sigma I), \rho = 1, \epsilon = 0, \gamma = \gamma_{\text{init}}$
1: Sample $E \sim M'$ and $z \sim \mathcal{N}_{\psi_0}$, reset environment
2: **for** $t \leftarrow 1$ to *num_prior_step* **do**
3:     With probability $\rho$, sample action $a_t \sim \pi^b(\cdot|o_t)$; else sample $a_t \sim \pi^p(\cdot|o_t, z)$
4:     With probability $\epsilon$, apply shielding $a_t^{\text{sh}} = \pi^{\text{sh}}(\pi^b, o_t, a_t)$
5:     Step environment $r_t, o_t, s_{t+1} \sim \mathcal{P}(\cdot|s_t, a_t^{\text{sh}})$
6:     Save $(o_{t+1}, o_t, a_t, a_t^{\text{sh}}, z, r_t)$ to replay buffer
7:     Update $\pi^p$ with reward and $\pi^b$ with DSBE
8:     Anneal $\rho \to 0, \epsilon \to 1, \gamma^b \to 1$
9:     **if** timeout or failure **then**
10:         Sample $E \sim \mathcal{M}'$ and $z \sim \mathcal{N}_{\psi_0}$, reset environment
11:     **end if**
12: **end for**
13: **return** $\pi^p, \pi^b, \mathcal{N}_{\psi_0}$

---

After the joint training, we obtain the dual policies $\pi^p$ and $\pi^b$, and the latent distribution $\mathcal{N}_{\psi_0}$ that encodes diverse trajectories in the environments. We now *fix* the weights of the two policies, and consider the latent variable $z$ also part of parameterization of the dual policy. This gives rise to the space of policies $\Pi := \{\pi_z^p, \pi^b : \mathcal{O} \mapsto \mathcal{A} \mid z \in \mathbb{R}^{n_z}\}$; hence, the latent distribution $\mathcal{N}_{\psi_0}$ can be equivalently viewed as a distribution on the space $\Pi$ of policies. In the next section, we will consider $\mathcal{N}_{\psi_0}$ as a *prior* distribution $P_0$ and "fine-tune" it by searching for a *posterior* distribution $P = \mathcal{N}_\psi$, which comes with the generalization guarantee from PAC-Bayes Control.

## 4.2 Safely Fine-Tuning Policies in Lab

In Lab stage we consider more safety-critical environments such as test tracks for autonomous cars or indoor lab space. After pre-training the performance and backup policies with shielding, the robot can safely explore and fine-tune the prior policy distribution $P_0$ in a new set of environments $\mathcal{M}$ sampled from the unknown distribution $D$. Leveraging the PAC-Bayes Control framework, we can provide "certificates" of generalization for the resulting posterior policy distribution $P$. The overall algorithm is similar to Algorithm 1. To avoid safety violations, we always apply value-based shielding to the proposed action ($\epsilon = 1$) during Lab training.

The PAC-Bayes generalization bound $R_{\text{PAC}}$ associated with $P$ from Eq. (1) consists of (1) $R_{\mathcal{M}}(P)$, the empirical reward of $P$ as the average expected reward across training environments in $\mathcal{M}$ (3), which can be optimized using SAC; (2) a regularizer $C(P, P_0)$ that penalizes the posterior $P$ from deviating from the prior $P_0$. Note that the only term in $C(P, P_0)$ that involves $P$ is the KL divergence term between $P$ and $P_0$. We modify the SAC formulation to include minimization of the KL divergence term. We consider stochasticity of the policy from the latent distribution instead of the policy network; this leads to removing the policy entropy regularization in SAC and adding a weighted KL divergence term to the actor loss. In practice, we find the gradient of the KL divergence term heavily dominates the noisy gradient of actor and critic, and thus we approximate the KL divergence with an expectation on the posterior:

$$\max_P \mathbb{E}_{o,z}\left[ \mathbb{E}_{a \sim \pi_\theta(\cdot|o,z)}\left[ Q^p(o, a) \right] - \alpha \log \frac{P(z)}{P_0(z)} \right]. \tag{7}$$

where $\alpha \in \mathbb{R}$ is a weighting coefficient to be tuned. After Lab training, we calculate the generalization bound $R_{\text{bound}}(P)$ using the posterior $P$. Please refer to Appendix A.1 for more details about the calculation. Overall, our approach provides generalization guarantees in novel environments from the distribution $D$: as policies are randomly sampled from the posterior $P$ and applied in test environments, the expected success rate over all test environments is guaranteed to be at least

$R_{\text{bound}}(P)$ (with probability $1 - \delta$ over the sampling of training environments; $\delta = 0.01$ for all experiments).

## 5  Experiments

We aim to answer the following in experiments: does our proposed Sim-to-Lab-to-Real achieve (1) lower safety violations during Lab training compared to other safe learning methods, (2) stronger generalization guarantees on performance and safety compared to previous work in PAC-Bayes Control, and (3) better empirical performance and safety during deployment compared to baselines?

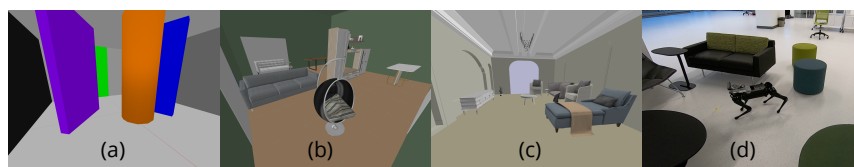

Figure 4: Samples of environments used in experiments: (a) Sim training in Vanilla-Env; (b) Sim training in Advanced-Env; (c) Advanced-Realistic training; (d) Real deployment with a quadrupedal robot.

**Environments.**  We evaluate the proposed methods by performing ego-vision navigation task in two types of environment. **Vanilla-Env** consists of undecorated rooms of $2m \times 2m$ with cylindrical and rectangular obstacles of different dimensions and poses, and the robot needs to reach a green door (Fig. 4a). A camera on the robot provides RGB images of $48 \times 48$ pixels. **Advanced-Env** uses the same room dimensions but places realistic furniture models from the 3D-FRONT dataset [24] (Fig. 4b). The robot needs to reach some target location using distance and relative bearing to the target. An onboard camera provides RGB images of $90 \times 160$ pixels.

In Sim training, we randomize obstacle and furniture configurations, and also camera poses (tilt and roll angles) in Advanced-Env to account for possible noise in real experiments. Sim training uses 100 environments in Vanilla-Env and 500 environments in Advanced-Env. After Sim training, we can fine-tune the policies in different types of Lab environments listed below:

- **Vanilla-Normal**: shares the same environment parameters as ones in the Sim stage.
- **Vanilla-Dynamics**: increases the lower bound of forward and angular velocity.
- **Vanilla-Task**: the robot needs to enter the target region with heading within some range.
- **Advanced-Dense**: assigns a higher density of furniture in the rooms.
- **Advanced-Realistic**: uses realistic room layouts (Fig. 4c) and associated furniture configurations from the 3D-FRONT dataset. We perform Lab-to-Real transfer with policies trained in this Lab (Fig. 4d). More details about the dataset can be found in Appendix A.3.

**Policy.**  We parameterize the performance and backup agents with NNs consisting of convolutional (CONV) layers and fully connected (FC) layers. The actor and critic of each agent share the same CONV layers. In Vanilla-Env, a single RGB image is fed to the CONV layers, and in Advanced-Env, we stack 4 previous RGB images while skipping 3 frames between two images to encode the past trajectory of the robot. More details of NN and training can be found in Appendix A.2.

**Baselines.**  We consider two variants of Sim-to-Lab-to-Real: **PAC_Shield** that trains a safety-aware policy distribution and **Shield** that trains a single safety-aware policy without conditioning on latent variables. We consider four types of baselines: (1) unconstrained RL that neglects safety violations (**Base**), (2) reward shaping that adds penalty to reward when violating constraints (**RP**), (3) PAC-Bayes control that trains a diverse policy distribution (**PAC_Base** and **PAC_RP** [8]), and (4) a separate safety agent (**SQRL** [6] and **Recovery RL** [7]). The major distinction between Sim-to-Lab-to-Real and PAC-Bayes control is that the latter does not handle the safety explicitly but only relies on diverse policies and fine-tuning to prevent unsafe maneuver. Sim-to-Lab-to-Real differs from SQRL and Recovery RL in that the latter train the safety critic with sparse safety indicators shown below,

$$Q^{\mathrm{b}}(o_t, a_t) := \mathcal{I}_E(s_t) + \gamma\big(1 - \mathcal{I}_E(s_t)\big) \min_{a_{t+1} \in \mathcal{A}} Q^{\mathrm{b}}\big(o_{t+1}, a_{t+1}\big),$$

where $\mathcal{I}_E(s_t) = \mathbb{1}\{g_E(s_t) > 0\}$ is the indicator function of the safety violations.

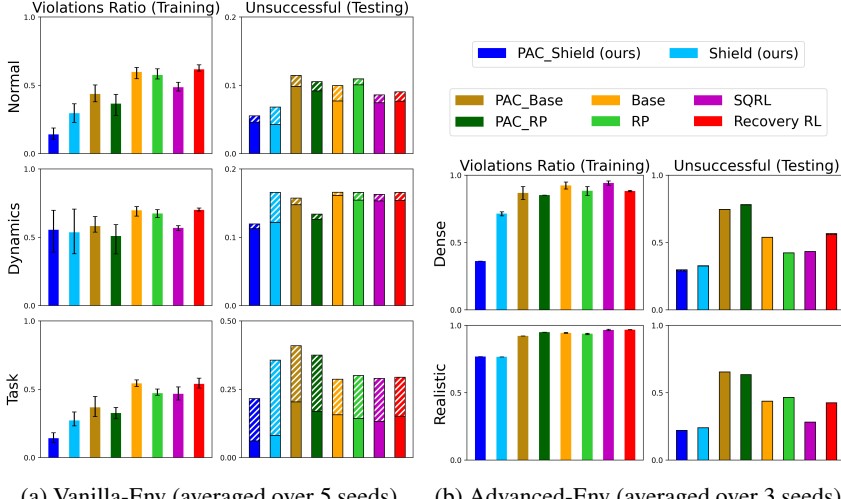

(a) Vanilla-Env (averaged over 5 seeds)  (b) Advanced-Env (averaged over 3 seeds)

Figure 5: Comparison of safety violations during Lab training and unsuccessful trials at test time.

**Results.** We compare all the methods by (1) safety violations in Lab training and (2) success and safety at deployment (Fig. 5). We calculate the ratio of number of safety violations to the number of episodes collected during training. For deployment, we show the percentage of failed trials (solid bars in Fig. 5) and unfinished trials (hatched bars). We summarize the main findings below:

1. Among all Lab training, our proposed Sim-to-Lab-to-Real (**PAC_Shield**) achieves fewest safety violations. This demonstrates the efficacy of using reachability-based safety critic for shielding as it learns from near failure with dense cost signals (as opposed to risk-based safety critics). Adding penalty in the reward function does not reduce safety violations significantly.

2. During deployment, Sim-to-Lab-to-Real achieves the lowest unsuccessful ratio of trajectories (solid bars plus hatched bars) and the fewest safety violations (solid bars). This suggests that (1) enforcing hard safety constraints explicitly improves the safety and (2) training a diverse and safe policy distribution achieves better generalization performance to novel environments. We show stronger generalization guarantees compared to PAC-Bayes baselines.

3. Sim-to-Lab-to-Real achieves the best performance and safety among baselines when the policies are deployed on a quadrupedal robot navigating through real indoor environments. The empirical performance and safety also validate the theoretical generalization guarantees.

**Reachability vs. Risk-Based Safety Critic.** Sim-to-Lab-to-Real and previous safe RL methods differ in (1) the metric used to quantify safety and (2) training of the backup agent. With reachability-based RL, we enforce the constraint that the *distance* to obstacles should be no lower than a threshold. In comparison, SQRL and Recovery RL define safety by the *risk* of colliding with obstacles in the future and use *binary* safety indicators. We argue that risk-based threshold can easily overfit to specific scenarios since the probability heavily depends on the discount factor used. In addition, reachability objective allows the backup agent to learn from near failure, while the risk critic in SQRL and Recovery RL needs to learn from complete failures, leading to more safety violations in Lab training.

Fig. 6 shows 2D slices of the safety critic values in both environment settings. Reachability-based critics output thicker unsafe regions next to obstacles, while risk critics fail to recognize many unsafe regions or consider unsafe only when very close to obstacles. Among different Lab setups, compared to the baselines, our method reduces safety violations by 77%, 4%, 76%, 62%, and 23% in training and 38%, 26%, 54%, 34%, and 28% in deployment. Through

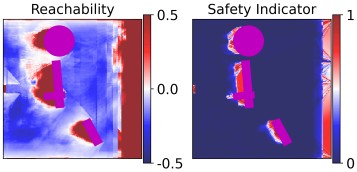

(a) Lab: Vanilla-Normal

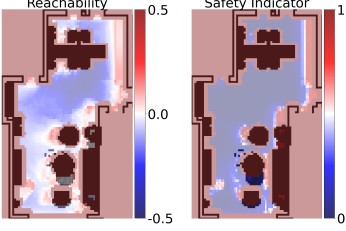

(b) Lab: Advanced-Realistic

Figure 6: 2D slices of safety critic values when the robot is facing to the right.

experiments we also find the value threshold $v_{thr}$ used in shielding an important parameter; see Appendix A.5 for more analysis.

**Generalization Guarantees.** We evaluate the PAC-Bayes generalization guarantees obtained after Lab training, and the effect of adding reachability-based shielding in the policy architecture to the bounds. Table 1 shows the bounds and test results on safety (not colliding with obstacles) and success (safely reaching the goal) in Advanced-Realistic Lab. The true expected success and safety are tested with environments that are similar to the Lab training environments (of the same distribution) but unseen before. We compare the bound trained using PAC_Shield with previous PAC-Bayes Control

Table 1: Results of PAC-Bayes guarantees and physical experiments with Advanced-Realistic Lab.

| Method | Advanced-Realistic | | |
|---|---|---|---|
| | PAC_Shield | PAC_Base | SQRL |
| # Lab Environments | 1000 | 1000 | 1000 |
| Success Bound | 0.701 | 0.297 | - |
| True Expected Success | 0.786 | 0.366 | 0.712 |
| Real Robot Success | 0.767 | 0.433 | 0.667 |
| Safety Bound | 0.708 | 0.304 | - |
| True Expected Safety | 0.794 | 0.367 | 0.713 |
| Real Robot Safety | 0.867 | 0.433 | 0.667 |

method (PAC_Base). With shielding, the bound improves from 0.366 to 0.786 for task completion and from 0.367 to 0.794 for safety satisfaction. Thus, explicitly enforcing hard safety constraints not only improves empirical outcomes but also provides stronger certification to policies in novel environments. Due to space constraint, we show the bounds for other Labs in Appendix A.4.

**Physical Experiments.** To demonstrate empirical performance and safety in real environments (Lab-to-Real transfer) and verify the generalization guarantees, we evaluate the policies in 10 real indoor environments with diverse layouts (see Appendix A.4 for more details). We deploy a Ghost Spirit quadrupedal robot equipped with a ZED 2 stereo camera at the front (Fig. 4d), matching the same dynamics and observation model used in Advanced-Realistic Lab. Before each trial, the robot is given the ground-truth distance and relative bearing to the goal at the initial location, and then it uses the localization algorithm native to the camera to update the two quantities.

We run policies trained with PAC_Shield (ours), PAC_Base (PAC-Bayes baseline), and SQRL (best overall among other baselines). Each policy is evaluated at one environment 3 times (30 trials total). The results are shown in Table. 1. Our policy is able to achieve the best performance (0.767) and safety (0.867), validating the theoretical guarantees from PAC-Bayes Control. The upper-right of Fig. 1 shows a trajectory when running policies trained with PAC_Shield in a kitchen environment.

# 6 Conclusion

We propose the Sim-to-Lab-to-Real framework that combines Hamilton-Jacobi reachability analysis and PAC-Bayes generalization guarantees to bridge the sim2real gap with a probabilistically guaranteed safety-aware policy distribution. We demonstrate significant reduction in safety violations in training and stronger performance and safety during test time.

**Discussion: Environment distribution.** As elaborated in Sec. 3, the generalization guarantees obtained through our framework assumes no distribution shift between Lab and Real in terms of environments. To bridge the discrepancy, we model the real environments by using (1) photorealistic dataset of indoor room layouts and furniture models and (2) dynamics from system identification of the real robot and camera poses. Additionally, we note that previous works in PAC-Bayes Control [8, 15, 16] have consistently shown real deployment validating the bounds. Even under a slight of shift in distribution, we believe that a certificate of performance and safety is useful and provides confidence for deploying the system.

**Discussion: Large-scale Lab training.** We acknowledge that one limitation of our framework is that, in exchange for assuming close to nothing about the environment distribution and providing statistical guarantees that hold in arbitrarily *high confidence* instead of in *expectation* only (e.g., conformal prediction [25]), we require at least a few hundred environments for "Lab" training to achieve tight PAC-Bayes generalization guarantees, which means performing "Lab" training with *real* conditions can be difficult for us researchers in university labs with limited resources. In this work, we resort to performing "Lab" training in realistic simulated environments. Nonetheless, we envision that our framework is well suited for industry practitioners who have access to either extensive training facilities (e.g. Google's robot "farms" [26] ), large-scale distributed systems (e.g. Amazon's warehouses [27]), or vast amounts of "Lab-like" data collection (e.g. Cruise and Waymo's thousands–millions of test driver miles [28]).

## Acknowledgments and Disclosure of Funding

Allen Z. Ren and Anirudha Majumdar were supported by the Toyota Research Institute (TRI), the NSF CAREER award [2044149], and the Office of Naval Research [N00014-21-1-2803]. This article solely reflects the opinions and conclusions of its authors and not ONR, NSF, TRI or any other Toyota entity. We would like to thank Zixu Zhang for his valuable advice on the setup of the physical experiments.

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

# A    Appendix

## A.1    Calculations of the PAC-Bayes Bound

After Lab training, we can calculate the generalization bound using the optimized posterior $P$. First, note that the empirical reward $R_S(P)$ involves an expectation over the posterior and thus cannot be computed in closed form. Instead, it can be estimated by sampling a large number of policies $z_1, ..., z_L$ from $P$: $\hat{R}_S(P) := \frac{1}{NL} \sum_{E \in S} \sum_{i=1}^{L} R(\pi_{z_i}^{p,b}; E)$, and the error due to finite sampling can be bounded using a sample convergence bound $\overline{R}_S$ [29]. The final bound $R_{\text{bound}}(P) \leq R_{\mathcal{D}}(P)$ is obtained from $\overline{R}_S$ and $C(P, P_0)$ by a slight tightening of $C_{\text{PAC}}$ from Theorem 1 using the KL-inverse function [8]. Please refer to Appendix A2 in [15] for detailed derivations.

## A.2    NN architecture and training details

We show the training hyperparameters used in Sim and Lab training in Table. A1 and Table. A2. In Vanilla-Env, the latent variable is appended to the output of the last CONV layer before FC layers. In

Advanced-Env, the stacked images are concatenated with the first 10 dimensions of the latent variable by repeating each dimension to the image size. Rest of the dimensions is appended to the output of the last CONV layer. The two auxiliary signals $\ell_E(s)$ and $\Delta_E(s)$ are also appended to the output of the last CONV layer.

Table A1: Hyperparameters for PAC_Shield in Sim training. Same NN architecture is used for performance and backup policies.

| | Environment Setting | | |
| --- | --- | --- | --- |
| | Vanilla-Normal/Dynamics | Vanilla-Task | Advanced-Env |
| # training steps | 500000 | 1000000 | 4000000 |
| Replay buffer size | 50000 (steps) | 100000 (steps) | 5000 (trajectories) |
| Optimize frequency | 2000 | 2000 | 20000 |
| # updater per optimize | 1000 | 1000 | 1000 |
| Value shielding threshold | -0.05 | -0.05 | -0.05 |
| **Latent Distribution** | | | |
| Latent dimension ($n_z$) | 20 | 20 | 30 |
| Augmented reward coefficient ($\beta$) | 2 | 2 | 2 |
| Prior standard deviation | 2 | 2 | 2 |
| **Optimization** | | | |
| Optimizer | Adam | Adam | Adam |
| Batch size (Performance) | 128 | 128 | 128 |
| Discount factor (Performance) | 0.99 | 0.99 | 0.99 |
| Learning rate (Performance) | 0.0001 | 0.0001 | 0.0001 |
| Batch size (Backup) | 128 | 128 | 128 |
| Discount factor (Backup) | $0.8 \to 0.999$ | $0.8 \to 0.999$ | $0.8 \to 0.99$ |
| Learning rate (Backup) | 0.0001 | 0.0001 | 0.001 |
| **NN Architecture** | | | |
| Input channels | 3 | 3 | 22[a] |
| CNN kernel size | [5,3,3] | [5,3,3] | [7,5,3] |
| CNN stride | [2,2,2] | [2,2,2] | [4,3,2] |
| CNN channel size | [8,16,32] | [8,16,32] | [16,32,64] |
| MLP dimensions | [130+$n_z$[b],128] | [132+$n_z$[b],128] | [248+$n_z$[b],256,256] |
| **Hardware Resource** | | | |
| # CPU threads | 8 | 8 | 16 |
| GPU | Nvidia V100 (16GB) | Nvidia V100 (16GB) | Nvidia A100 (40GB) |
| Runtime | 8 hours | 14 hours | 12 hours |

[a] We stack 4 previous RGB images while skipping 3 frames between two images and concatenate the stacked images with the first 10 elements of the latent variable (each element is repeated to match the same shape of a channel in an image).

[b] The input of the first linear layer is composed of the output from the convolutional layers, latent variables and auxiliary signals, which is $128 + n_z + 2$ in Vanilla-Normal/Dynamics, $128 + n_z + 4$ in Vanilla-Task and $256 + (n_z - 10) + 2$ in Advanced-Env.

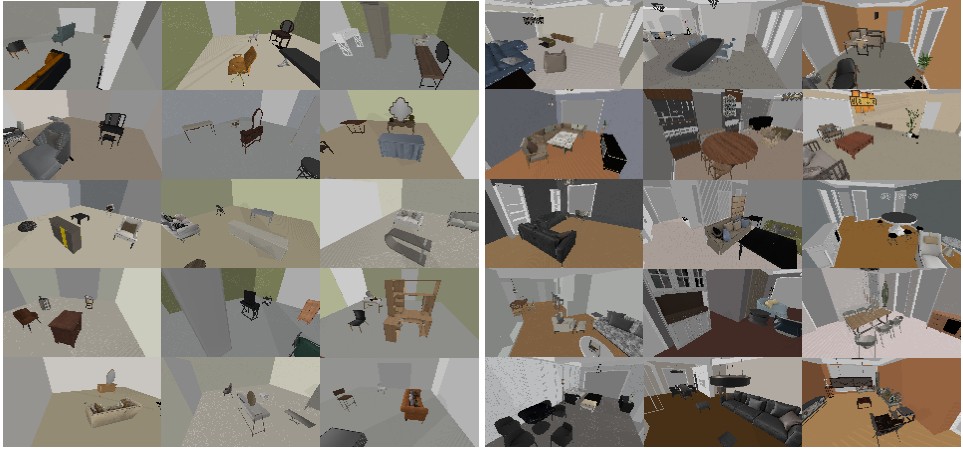

(a) Sim training          (b) Lab training

Figure A1: **Samples of robot observations in Advanced-Env**: for better view here, the virtual camera is placed at a higher location than the robot.

Table A2: Hyperparameters for PAC_Shield in Lab training.

| | Environment Setting | |
|---|---|---|
| | Vanilla-Env | Advanced-Env |
| # training steps | 500000 | 3000000 |
| Replay buffer size | 50000 (steps) | 5000 (trajectories) |
| Optimize frequency | 2000 | 20000 |
| # updater per optimize | 1000 | 1000 |
| Value shielding threshold | -0.05 | -0.05 |
| The number of environments ($N$) | 1000 | 1000 |
| **Optimization** | | |
| Learning rate for latent mean | 0.0001 | 0.0001 |
| Learning rate for latent std | 0.0001 | 0.0001 |
| KL-divergence coefficient ($\alpha$) | 1 | 2 |
| Optimizer | Adam | Adam |
| Batch size (Performance) | 1024 | 128 |
| Discount factor (Performance) | 0.99 | 0.99 |
| Learning rate (Performance) | 0.0001 | 0.0001 |
| **PAC-Bayes Bound** | | |
| The number of latent variables ($L$) | 1000 | 1000 |
| Precision ($\delta$) | 0.01 | 0.01 |
| **Hardware Resource** | | |
| # CPU threads | 8 | 8 |
| GPU | Nvidia V100 (16GB) | Nvidia A100 (40GB) |
| Runtime | 6 hours | 16 hours |

## A.3 Environment Setup for Advanced-Env

In order to train the navigating agent in realistic environments before Real deployment, we use the 3D-FRONT dataset [24] that offers a larger number of synthetic indoor scenes with professionally designed layouts and high-quality textured furniture. This is the richest dataset we find suitable to indoor navigation task, as training with domain randomization and PAC-Bayes Control framework often requires more than 1000 environments.

For Sim training, we use $7m \times 7m$ undecorated rooms as room layouts, and randomly placing 5 pieces of furniture from the dataset. We use 4 categories of furniture: Soft (2701 pieces available), Chair (1775 pieces), Cabinet/Shelf/Desk (5725 pieces), Table (1090 pieces). We also randomly sample textures from the dataset to add to the walls and floor: for walls, we use categories Tile, Wallpaper, and Paint (911 images available in total), and for floor, we use Flooring, Stone, Wood, Marble, Solid Wood Flooring (466 images). We set the minimum clearance between furniture, around the initial location, and around the goal to be $1m$. The minimum distance between the initial location and the goal is $5m$. Fig. A1(left) shows samples of observations at the initial locations. For Advanced-Dense Lab where the furniture density is higher, we place 6 instead of 5 pieces of furniture, and the minimum clearance is $0.8m$ instead of $1m$.

For Lab training, we instead use the professionally designed room layouts (with furniture configuration) from the dataset. The dataset contains 6813 different house layouts (each with multiple rooms). Since our focus is on obstacle avoidance with relatively short horizon, in each house, we sample initial and goal locations within one room. Unfortunately the dataset does not provide corresponding wall and floor textures in each layout, and we resort to random samples as in Vanilla-Env. Again we maintain a minimum clearance of $1m$ between furniture, around the initial and goal locations. To check the environment is solvable, we extract a 2D occupancy map for each room and run the Dijkstra algorithm. We also ensure there is at least one piece of furniture along the line connecting the initial and goal locations. At the end, we process 2000 room environments, which are then split for training and testing. Fig. A1(right) shows samples of observations at the initial locations.

## A.4 Supplementary Experiment Results

We present the PAC-Bayes bound for different labs in Table. A3. For all labs, explicitly handling safety constraints with shielding improves the performance and safety bound as well as the empirical results. Fig. A2 shows the 10 real environments and robots' trajectories when running policies trained with PAC_Shield. The first and third images on top of the figure show the robot's view when shielding

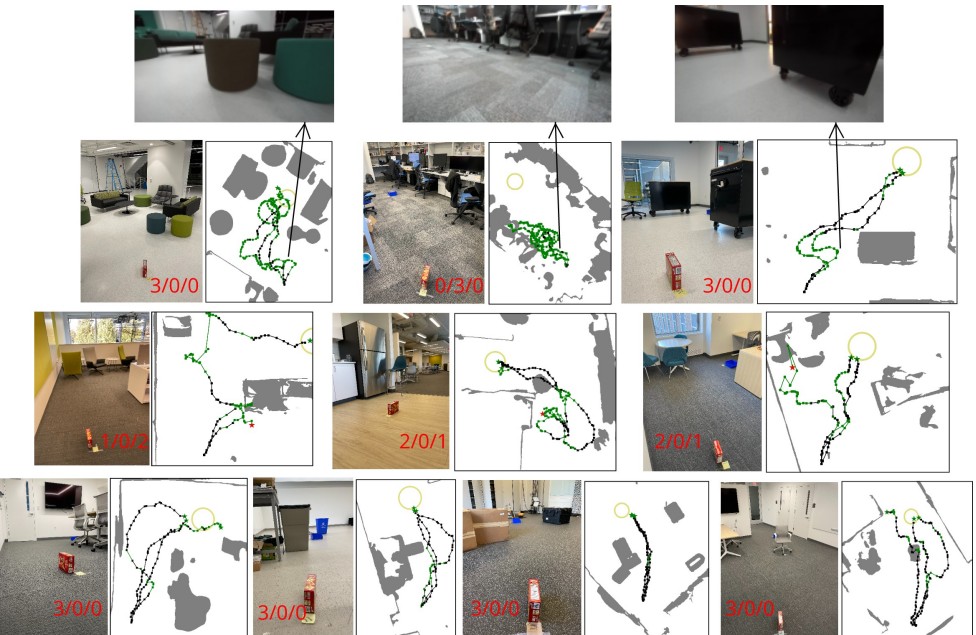

Figure A2: **Environments for physical robot experiments and robot trajectories/observations with PAC_Shield:** we run the policy three times in each environment by sampling different latent variables from the posterior distribution. The three numbers in images indicates success/unfinished/failure split. Green dots indicates shielding in effect. Green star indicates success in reaching the target. Red star indicates colliding with obstacles. We scan the environment using an iPad Pro tablet before experiments to generate the 2D map. The robot trajectory is obtained using localization algorithm of the onboard camera, and is inaccurate at places (intersecting obstacles; not exactly reaching the target but the robot deems so, which we consider success).

Table A3: PAC-Bayes bound in different Labs.

| | Vanilla-Normal | | Vanilla-Dynamics | | Vanilla-Task | |
|---|---|---|---|---|---|---|
| Method | PAC_Shield | PAC_Base | PAC_Shield | PAC_Base | PAC_Shield | PAC_Base |
| # Lab Environments | 1000 | 1000 | 1000 | 1000 | 1000 | 1000 |
| Success Bound | 0.876 | 0.735 | 0.820 | 0.778 | 0.757 | 0.468 |
| True Expected Success | 0.945 | 0.886 | 0.880 | 0.843 | 0.851 | 0.590 |
| Safety Bound | 0.911 | 0.816 | 0.835 | 0.815 | 0.884 | 0.663 |
| True Expected Safety | 0.954 | 0.902 | 0.887 | 0.852 | 0.939 | 0.796 |

| | Advanced-Dense | | Advanced-Realistic | |
|---|---|---|---|---|
| Method | PAC_Shield | PAC_Base | PAC_Shield | PAC_Base |
| # Lab Environments | 1000 | 1000 | 1000 | 1000 |
| Success Bound | 0.623 | 0.254 | 0.701 | 0.297 |
| True Expected Success | 0.703 | 0.327 | 0.786 | 0.366 |
| Safety Bound | 0.630 | 0.259 | 0.708 | 0.304 |
| True Expected Safety | 0.709 | 0.332 | 0.794 | 0.367 |

successfully guides robot away from the sofa stool and the cabinet. In the second environment, the backup policy keeps shielding the robot away from center of the room with $v_{thr} = -0.10$, and all three trials ended as unfinished. We also test with $v_{thr} = -0.05$, and the robot is able to reach the target without shielding always activated. This highlights the need for adapting the shielding value threshold online in future work.

## A.5 Other Studies

**Ablation Study: importance of two-stage training** We evaluate the significance of Lab training by testing the prior policy distribution (without fine-tuning in Lab) in Vanilla-Env. Without Lab training, the unsuccessful ratio in deployment increases by $16\%$, $8\%$ and $14\%$. This suggests that Lab training is essential to policies adapting to real dynamics and new distribution of environments.

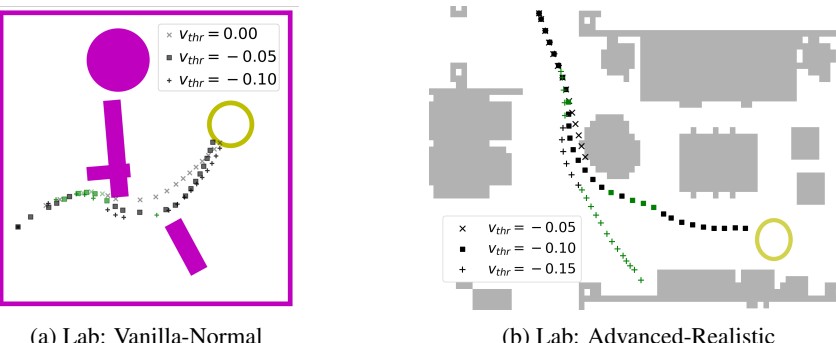

(a) Lab: Vanilla-Normal          (b) Lab: Advanced-Realistic

Figure A3: **Rollout trajectories using different value threshold for shielding:** higher threshold (more negative) results in more conservative maneuver, i.e., farther away from obstacles. In Advanced-Env, we tend to find too high threshold prevents the robot from reaching the goal and accidentally steers it towards tight space.

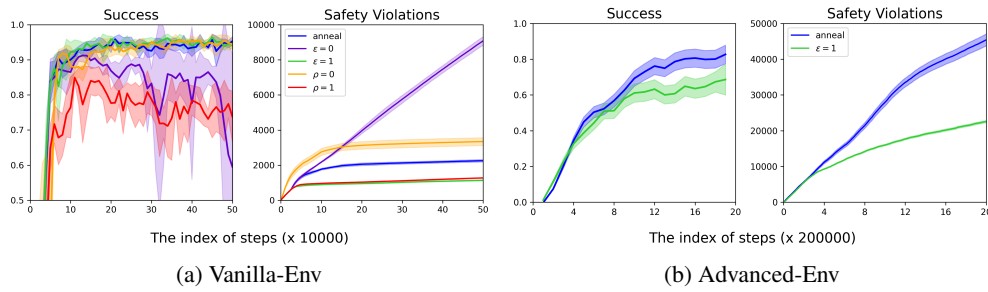

(a) Vanilla-Env                  (b) Advanced-Env

Figure A4: **Effect of $\rho$ and $\epsilon$ scheduling in Sim training:** annealing $\rho$ and $\epsilon$ helps balance between safety violations and task completion. For Vanilla-Env, $\rho$ initializes at $1$ and decays by $0.5$ every $25000$ steps, and $\epsilon$ initializes at $0$ with $1 - \epsilon$ decaying by $0.5$ every $50000$ steps. For Advanced-Env, $\rho$ initializes at $0.5$ and decays by $0.5$ every $500000$ steps, and $\epsilon$ initializes at $0$ with $1 - \epsilon$ decaying by $0.5$ every $200000$ steps.

Additionally, we test the importance of Sim training with the baseline *Shield* (no policy distribution). Without Sim training, the safety violations in Lab training increases by $60\%$, $11\%$ and $65\%$. This demonstrates that Sim training enables the backup agent to monitor and override unsafe behavior from the beginning of Lab training.

**Sensitivity analysis: value threshold**    Through experiments, we find the value threshold used in shielding essential to performance and safety. $v_{thr} = 0$ naturally results in more safety violations during training compared to $v_{thr} = -0.05$ and $v_{thr} = -0.10$. Policies trained with $v_{thr} = 0$ also performs the worst at test time, which indicates that less shielding during training makes the robot learn unsafe or aggressive maneuver. Next we evaluate how the value threshold affects robot trajectories at *test* time. Fig. A3 shows the trajectories using different thresholds in the two settings. Small threshold leads to robot passing very closely next to obstacles, while a bigger threshold leads to more conservative behavior. We also would like to highlight the challenges of learning safe policies in Advanced-Env. As shown in the figure, with $v_{thr} = -0.15$ the robot avoids the first obstacle, and then the backup policy steers the robot away from the target, potentially deeming the clearance next to the target not sufficient. However, this brings the robot near the wall, and due to imperfect training of the backup actor, the robot fails to escape. With tight spacing and large dimensions of the robot in Advanced-Env, we find the backup agent more difficult to train, and the final test performance and safety can be sensitive to the shielding threshold. In Advanced-Realistic, average test success rate with $v_{thr} = -0.05, -0.1, -0.15$ are 0.678, 0.786, and 0.762 respectively. Future work could look into adapting the threshold after short experiences in different environments.

**Sensitivity analysis: the probability of sampling actions from the backup policy ($\rho$) and the probability of activating shielding ($\epsilon$)**    One of the main contributions of our work is the effective joint training of both performance and back agents (realized in Sim training). The two parameters, $\rho$

and $\epsilon$, directly affect the exploration in Sim training. With high $\rho$ or high $\epsilon$, the RL agent basically only explores conservatively within a small safe region. However, in the beginning of the training, we should allow the RL agent to collect diverse state-action pairs. On the other hand, we also gradually anneal $\rho \rightarrow 0$ and $\epsilon \rightarrow 1$ since we want the performance policy to be aware of the backup policy. In other words, the performance policy is effectively in *shielded environments* towards end of Sim training. Fig. A4 shows the Sim training progress under different $\rho$ and $\epsilon$ scheduling. With constant $\rho = 0$ or $\epsilon = 0$, the number of safety violations is much higher than that with both parameters annealing. Even worse, $\epsilon = 0$ results in the number of safety violations increase at constant speed and the training success fluctuates significantly. On the other hand, with $\rho = 1$ or $\epsilon = 1$, the number of safety violations is only half as that with both parameters annealing. However, this is at the expense of exploration and leads to worse success rate in deployment. In Vanilla-Env $\rho = 1$ leads to very poor training success. Although in Vanilla-Env $\epsilon = 1$ does not have significant effect on training success, in the Advanced-Env, insufficient exploration hinders training progress. Also note that Sim training is not safety-critical and we do not aim to reduce safety violations then.

