# OpenReview forum: "Sim-to-Lab-to-Real: Safe Reinforcement Learning with Shielding and Generalization Guarantees"
_NeurIPS.cc/2022/Workshop/TEA — TEA_

### Official Review · Reviewer_FRq8 · 2022-10-11
**I recommend that this work be accepted.**

**Rating:** 8
**Confidence:** 3

**Review:**

## Summary

This work presents a PAC-Bayes-based learning paradigm that utilizes an intermediary step called "Lab" to bridge the Sim-to-Real gap. This allows the authors to generate probabilistic guarantees on the success and safety of the robot's actions while also empirically improving upon other baseline methods. The authors provide a thorough comparison of their method against existing baselines and provide significant discussion of parameter sensitivity in the appendix. They also present a compelling hardware demo of their method.

---

## Quality

This paper appears to be of high quality.

---


## Clarity

The paper is generally clear, but could use some improvement.

Some unclear portions include:
- The differentiation between the ***Sim*** and ***Lab*** stages is confusing in the Experiments section since the ***Lab*** portion is also simulated and not performed on a "test track" as suggested in the Introduction. The authors mention the reason for this in the Conclusion, but this comes significantly after the description of their experiments. I would suggest moving this discussion to the Environments subsection of Section 5.
- In the results section, it is mentioned that the "ratio of number of safety violations to the number of episodes collected during training" is calculated; however, the number of episodes is not directly mentioned here or in Table A.1. Furthemore, in Appendix A.3, it is mentioned that the large number of environments required to utilize the PAC-Bayes Control framework is a limitation. This limitation is obfuscated by the averaging performed to obtain the results in Table A.1 and should be made more clear.
- Appendix A.5's **Ablation study** mentions that "the unsuccessful ration in deployment increases by 16%, 8%, and 14%". What are these three separate numbers referring to?
- The text in Figure 3 is too small to read.
- The notation $s \in \mathcal{S} \subseteq \mathbb{R}^{n_s}$ and $a\in \mathcal{A} \subseteq \mathbb{R}^{n_a}$ is clear, but the set $\mathcal{E}$ is not similarly defined. Can you further describe this set?
- There is some lack of clarity when the function $g$ is introduced. Is $g$ Lipschitz in both of its arguments or just in $s$? And is $g_E(s) \triangleq g(s,E)$?
- It is mentioned that the robot is considered to have discrete-time dynamics $s_{t+1} = f_E(s_t, a_t)$. What are these dynamics for robot used in the experimental section? Are the torques of the quadrupeds joints being controlled or is a multi-level control structure utilized? What is the time step used in the discrete dynamics? Furthermore, the authors mention that the same dynamics used in the physical experiment are "used in Advanced-Realistic Lab". Other than the altered velocity bounds, are the same dynamics used for all of the other *Sim* and *Lab* environments?
- Figure 1 is introduced in the text in paragraph 2 of the introduction, but contains a large amount of notation that is not introduced until Section 3. It would improve readability of this figure if the math notation were replaced with simplified text that the reader can understand while reading the introduction.

Minor typos include:
- in the abstract: "[...] policy distribution **..**"
- "Domain randomization [...] **do** not explicitly adress safety of the robot"
- "[...] when introducing the rest of formulation and the approach [...]"
- I believe that "Maximizing the lower bound $R_\textrm{PAC}$ can be viewed as maximizing the empirical reward $R_\mathcal{M}(P)$ along with a regularizer $C$ that prevents overfitting by penalizing the deviation of the posterior $P$ from the prior $P_0$." should probably be "Maximizing the lower bound $R_\textrm{PAC}$ can be viewed as maximizing the empirical reward $R_\mathcal{M}(P)$ along with ***minimizing*** a regularizer $C$ that prevents overfitting by penalizing the deviation of the posterior $P$ from the prior $P_0$"
- "With reachability-based RL, **we enforces** [...]"



---
## Originality

The authors extend the existing PAC-Bayes baseline. The theoretical analysis that additional learning performed on "Labs" sampled from the distribution $D$ would improve would improve performance in a "Real" world sampled from $D$ is straightforward, but the work provides thorough and novel exploration of this idea.


---

## Significance of this Work

This work addresses the highly relevant problem of navigating safely in novel environments. Although a significant number of safety violations occur, the authors show that they significantly improve on existing methods.

The proposed method appears to be limited by the fact that:
1. the *Real* environments must be drawn from the same distribution as the *Lab* environments, limiting generalizability, and
2. a very large number of *Lab* environments are needed to generate useful bounds when training with domain randomization and the PAC-Bayes Control framework (as mentioned in A.3).

---


## List of Pros and Cons

### Pros:
 - The paper generates probablistic theoretical bounds for safety of learned controllers.
 - The proposed method is shown to outperform a variety of other baseline methods.
 - A significant amount of information regarding the learning framework is provided in the appendix which helps improve reproducibility of the result.

### Cons:
- The presentation is occasionally unclear.
- The generalization guarantees assume no distribution shift between *Lab* and *Real* environments, limiting generalizability of the method and its guarantees.
- A large number of *Lab* environments are required when training with a PAC-Bayes Control framework.
- No code or data are provided. Access to the code and data used to produce the results in Figures 5 would significantly improve the reproducibility of this result.

---

### Official Review · Reviewer_C523 · 2022-10-15
**Interesting Combination of Hamilton-Jacobi reachability analysis an PAC-Bayes control**

**Rating:** 7
**Confidence:** 3

**Review:**

The paper adds an intermediate stage "lab" in the sim-real transfer and provides performance and safety guarantees for robots trained in this framework. A distribution of policies is learned in the "sim" setting which is then fine-tuned in the "lab" setting before deploying in the "real" setting. These distributions are treated as the prior and posterior respectively to obtain PAC-Bayes bounds on performance. Safety is ensured by applying Hamilton-Jacobi reachability analysis to train a backup policy that takes over when it predicts the original policy to violate safety constraints.

Strengths:
* I think the idea of combining reachability analysis and PAC-Bayes bound is an interesting way to provide theoretical bounds for safe RL.
* Experiments are well-thought-out and I think they provide justification for the claims made in the paper.
* The theoretical guarantees transferring to actual experiments is nice to see.

Weaknesses:
* My main concern is the assumption that "lab" and "real" environments come from the same distribution. The authors justify it (line 373) by saying that "lab" environments were modeled to be close to "real" environments and minor distribution shifts wouldn't affect the performance much. But this might be very difficult in certain settings, e.g., in the self-driving example given by the authors since accurately modeling the behavior of pedestrians and other cars is a difficult problem.
* If there is a gap between "lab" and "real", I think the value-based shielding (eq. 6) will be less reliable. Additionally, the value threshold for switching the policy will also be hard to tune. I think further study is necessary to ensure that it is not too detrimental to the performance or safety of the agent in "real" environments.
* The policy training seems sensitive to the annealing schedules of $\rho$ and $\epsilon$, so I wonder if it would be difficult to set good schedules when applying this method to different problems.

Misc:
* I would be interested to see how the theoretical guarantees can be extended to cases in which the weights of the policy network are also fine-tuned in the "lab" phase instead of just the distribution of latent vectors.

Minor:
* I found the beginning of section 4.1 hard to follow since the Q-value and the Bellman equation are different from the standard ones used in RL. I think an intuitive description of the equation, similar to how it is done in "Bridging Hamilton-Jacobi Safety Analysis and Reinforcement Learning" by Fisac et al., would be helpful.

Overall, I would say that the framework introduced in this paper is interesting enough to warrant further discussion, and hence, I recommend the paper be accepted.

Disclaimer: I am not very familiar with the PAC-Bayes control framework and hence, I have not verified the math behind the bounds given in the paper.

---

### Official Review · Reviewer_oaYd · 2022-10-18

**Rating:** 7
**Confidence:** 3

**Review:**

The authors propose a framework for learning an RL policy that performs well in the real and enjoys performance and safety guarantees. To achieve this, this paper proposes a framework called Sim-to-Lab-to-Real which combines Hamilton-Jacobi reachability analysis and PAC-Bayes generalization guarantees.

In the "sim" stage, a performance and a backup policy are jointly trained using RL. In the "lab" stage, the generalization bounds are optimized to certify the robot's performance and safety before deployment. Results from numerical and physical experiments show that the proposed method can effectively reduce safety violations.

Generally speaking, this framework generalizes the popular sim-to-real scheme, and it is a fascinating idea. This paper is also very relevant to this workshop's agenda: trustworthy embodied AI. The writing is good and experimental results are promising.

My main criticisms are:
(1) What is the definition of "safety" here? It seems that this paper cannot guarantee full-state safety (because the minimax problem is in general intractable).
(2) The PAC-Bayes perspective is very interesting. However, it is unclear in which case the PAC-Bayes bound will be vacuous. If the reviewer understands correctly, the PAC-Bayes bound gives an average performance guarantee. Is it still suitable for the worst-case scenario?

---

### Decision · Program_Chairs · 2022-10-21

**Decision:**

Accept

**Comment:**

The paper is well-written and of high quality. The proposed method is well-motivated and throughoutly justified both theorectically and empirically. Please consider the comments from the reviewers in the final version.